# Can Methotrexate Be Employed as Monotherapy for Bullous Pemphigoid? Analysis of Efficiency and Tolerance of Methotrexate Treatment in Patients with Bullous Pemphigoid

**DOI:** 10.3390/jcm12041638

**Published:** 2023-02-18

**Authors:** Magdalena Wojtczak, Amanda Nolbrzak, Anna Woźniacka, Agnieszka Żebrowska

**Affiliations:** Department of Dermatology and Venereology, Medical University of Lodz, 90-647 Lodz, Poland

**Keywords:** bullous pemphigoid, methotrexate, immunosupressive therapy, glucocorticosteroids

## Abstract

The European Academy of Dermatology and Venerology (EADV) consensus states that the treatment of choice for bullous pemphigoid is systemic glucocorticosteroid therapy. Bearing in mind that long-term steroid therapy is associated with numerous side effects, an effective and safer treatment regimen for these patients is still being sought. A retrospective analysis was performed of the medical reports of patients with diagnosed bullous pemphigoid. The study included 40 patients with moderate or severe disease, and who had continued ambulatory treatment for at least six months. The patients were divided into two groups: one treated with methotrexate in monotherapy, or with combined methotrexate and systemic steroid therapy. A slightly better survival rate was noted in the methotrexate group. No significant differences were observed between the groups in time to achieve clinical remission. The combination therapy group demonstrated more frequent disease recurrence and exacerbations during treatment, and a higher mortality rate. None of the patients in either group presented with severe side effects related to methotrexate treatment. The treatment of bullous pemphigoid with methotrexate in monotherapy is an effective and safe therapeutic method for elderly patients.

## 1. Introduction

Bullous pemphigoid (BP) is an autoimmune blistering disease characterized by the presence of bullous lesions localized in the skin and/or mucous membrane. BP commonly affects the elderly, primarily those above the age of 65. BP is a chronic disease with a tendency to recur. The European Academy of Dermatology and Venerology (EADV) consensus states that the treatment of choice for BP is systemic glucocorticosteroid therapy [1]. However, as steroid therapy may result in steroid-related systemic effects in patients with BP, alternative methods of treatment are still being proposed. As such, there is great interest in other regimens, including those implementing immunosuppressants, such as methotrexate, to find the optimal method that minimizes the side effects of systemic glucocorticosteroid [2,3,4,5,6].

### 1.1. Disease Overview

BP is a common autoimmune blistering disease. While it is most frequently observed in elderly patients, it may appear at any age. Even so, it is estimated that the risk of developing BP is 300 times greater at 90 years old than at 60 years old. Previous studies indicate that the presence of class II DQB1*0301 HLA antigens is associated with a higher chance of developing BP [7].

The clinical symptoms of BP, especially in the early stages of the disease, are diverse and mostly non-specific. They can manifest as prurigo-like lesions, eczematous lesions or urticarial lesions on the skin [1]. Over time, patients develop tense serous-filled bullous lesions, which can arise on healthy skin, erythematous skin or on erosions caused by ruptured blisters. Despite this varied clinical picture, the most common complaint reported by patients is the itching of the skin. Isolated pruritus may be the first symptom of the disease [8]. In approximately 10–30% cases, the disease affects the mucosa, taking the form of classic bullous lesions or painful, demarcated erosions without any tendency to leave scarring. These skin eruptions mostly appear on the buccal mucosa [9].

According to the BPDAI scale, BP can manifest as mild, moderate or severe. BP treatment is long term, and usually has to be modified to maintain remission. The mainstay of treatment for BP is based on systemic glucocorticosteroids. The EADV consensus recommends prednisone, 0.5–0.75 mg/kg per day. As high-dose systemic steroid therapy, such as prednisone 1 mg/kg per day, is associated with higher mortality and increased side-effects, this dosage is not recommended during the initial treatment. In contrast, doses lower than 0.5 mg/kg per day seem to be less effective in achieving remission, giving unsatisfactory results and requiring increased doses or the addition of another immunosuppressive drug [1].

It is recommended to combine systemic and topical steroid therapy. After the patient achieves complete remission, the treatment may be discontinued after three to six months of regular therapy. However, expert opinions vary, and this regimen has not yet been validated. Therefore, to reach remission and maintain the good condition of the patient, the total treatment duration is usually extended to nine to twelve months [1].

A greater risk of side effects, such as strokes, thromboembolic events, diabetes, hypertension, psychosis and osteoporosis, has been reported in older patients [10,11,12], in addition to continuously looking for the optimal strategy for therapy. Considering the side effects of steroid therapy and the risk of mortality in older patients during long-term steroid therapy, methotrexate (MTX) seems to be a good alternative treatment in elderly patients with BP.

Methotrexate is an antimetabolic drug that acts as an antagonist of folate. It inhibits dihydrofolate reductase, an enzyme that catalyzes the conversion of dihydrofolate to tetrahydrofolate, resulting in a deficiency of tetrahydrofolate, an active form of folate. It hinders the synthesis of purine bases and the nucleoside thymidine required for DNA synthesis and the conversion of homocysteine into methionine. Methotrexate affects cell proliferation, especially the S phase of the cell cycle. MTX is eliminated mainly by renal excretion; approximately 50–80% of the drug is eliminated in an unchanged form during the first 12 h. In addition to renal function, MTX elimination is also influenced by age, body weight, race, the existence of leaks into body cavities, hydration level and the use of other drugs [13,14]. A decrease in MTX may cause side effects, including bone marrow damage, renal or liver dysfunction, gastrointestinal symptoms, skin rashes, neurotoxic symptoms and increased risk of infection [15].

In dermatology, MTX has been used to treat inter alia psoriasis, psoriatic arthritis, bullous diseases, atopic dermatitis, eczema, lichen planus, pityriasis rubra pilaris, chronic urticaria, scleroderma, dermatomyositis and lupus erythematosus [16]. Some papers have examined the use of MTX in the treatment of pemphigoid or pemphigus. [17,18]

Furthermore, it has been reported that the efficiency of the MTX treatment and frequency of side effects varies considerably between patients. Therefore, there is great interest in identifying biochemical and clinical markers, which may predict the response to MTX treatment. Recent studies have been paying more attention to the role of genetic factors that may influence variation in the activity of the cell surface receptors for MTX [19,20].

### 1.2. Objectives

The main aim of this study was to analyze the efficiency of MTX treatment and tolerance to it. It compares the effects of MTX monotherapy with a combined regimen of MTX and systemic steroid therapy. Due to the high risk of side effects in elderly patients treated with long-term high doses of steroids, no patients were treated only with systemic steroid therapy at the doses recommended by the EADV consensus.

## 2. Materials and Methods

A retrospective analysis was performed of the medical reports recorded at the Department of Dermatology and Venerology and Ambulatory Care Center, Medical University of Lodz between 2009 and 2019. The study was approved by the Ethics Committee of the Medical University of Lodz.

In total, 120 patients with bullous pemphigoid diagnosed on the basis of immunopathological examination were enrolled in the study. Patients who did not achieve treatment compliance and stopped taking methotrexate were excluded; MTX was discontinued due to high liver test values and severe anemia. In addition, patients with advanced cancer were also excluded.

Ultimately, 40 patients with BP were included in the retrospective analysis, all of whom continued the ambulatory treatment for at least six months. Patients were monitored for adverse events throughout the treatment period and after the treatment discontinuation. The shortest follow-up time was six months, but this was extended if deemed necessary.

In pemphigoid patients, the standard follow-up in terms of survival parameters was three years.

Selected patients included in the study had a moderate or severe type of BP. The medical records were reviewed while performing the retrospective analysis. The advancement of the illness was scored based on the BPDAI (BP disease area index) scale. Clinical remission was defined as the absence of new active skin lesions, such as erythema, vesicles and blisters on the skin and/or the mucous membranes.

In our department, mild BP is managed with topical steroid therapy: clobetasol propionate cream according to Joly’s method. Therefore, it was not included in the study [12].

The remaining patients with symptoms of BP were treated with MTX in monotherapy or a combined regimen of the MTX and systemic steroid therapy.

### Statistical Methods

The experimental data were collected in MS Excel. The mean value and standard deviation of the studied population was calculated. The qualitative data were analyzed using the **χ**^2^ test, and the quantitative data using Student’s *t*-test. Additionally, Pearson’s correlation coefficient (r) was determined. Differences were regarded as significant at a level of *p* < 0.05. All analyses were performed using Statistica 13.1 software.

Survival analysis was performed by dividing overall survival after diagnosis by the survival observed in a similar population not diagnosed with the disease. The latter population is composed of individuals with at least similar age and gender to the population with the disease.

The mortality ratio indicates the ratio of the number of deaths in a given population to the number of expected deaths; a form of indirectly standardized rates. The categories are usually defined by age, gender, population type or ethnicity.

## 3. Results

The 40 studied patients with BP treated in our department were suffering from moderate to severe forms of the disease: 38 ± 8 according to the BPDAI scale.

Pemphigoid was diagnosed based on the clinical symptoms together with the histological and immunological findings. In all cases, the histopathologic findings, according to the Ackerman scale, were fully developed. Direct immunofluorescence testing revealed IgG/C3 linear deposits along the BMZ in all the patients. In the salt split test, deposits were observed in the epidermal part of the blister. Circulating anti-BMZ IgG antibodies were detected in 13 out of the 40 patients, using an indirect immunofluorescence test (titer 1:40–1:320) (Euroimmun, Germany). In addition, anti-NC16a autoantibodies were present in the serum of 29 out of the 40 patients, as assessed by an ELISA test (MBL, Nagoya, Japan).

The patients were divided based on the chosen therapy: Group 1 comprising patients treated with MTX in monotherapy and Group 2 comprising those treated with a combination of MTX and systemic steroid therapy. The mean patient age was 77.5 ± 11.54 years. Topical steroid therapy for skin lesions was implemented in all groups as additional treatment.

### 3.1. Group 1

In Group 1 (MTX in monotherapy), the mean patient age was 81.75 ± 10.04 years. The patients began MTX treatment with a starting weekly dosage of 7.63 ± 1.67 mg (range 5–10 mg). Following this, 55% of the patients received a dose of 7.5 mg/week, 25%—10 mg/week and the remaining 20% received 5 mg/week. In 25% of the patients, the weekly dosage of MTX was increased to a mean value of 10.50 ± 2.45 mg/week due to failure to achieve remission (range 7.5–15 mg). The minimum dose of MTX in clinical remission was 4.88 ± 2.32 mg/week (range 2–10 mg).

The clinical remission rate was 100%. The mean time to remission was four months of treatment (3.80 ± 3.17). The survival rate was 100%.

All of the patients suffered from lesions localized on the skin. In addition, the mucous membranes were affected in 5% of cases. MTX therapy was not influenced by the history of internal diseases, such as hypertension or diabetes, or by treatment with furosemide or neuroleptics.

In 25% of the patients, MTX was withdrawn; clinical remission was maintained in 60% of these cases. Attempts were made to discontinue MTX treatment after 15.90 ± 6.05 months. In Group 1, the MTX therapy was well tolerated in 100% of cases and there was no need to withdraw MTX due to side effects. The main complaints reported by the patients were hair loss (in 5%) and weakness and fatigue (in 10%).

Laboratory tests (LFTs, morphology) conducted during the therapy showed no deviations from the norm. Otherwise, 10% of the patients demonstrated longer periods to clinical remission due to noncompliance with medication, and 42.1% presented with an exacerbation of the disease while reducing MTX doses.

### 3.2. Group 2

In Group 2 (MTX + GKS), the mean patient age was 73.35 ± 11.42 years. The mean prednisone dose used to achieve remission was 21.58 ± 12.68 mg (range 10–60 mg). In 40% of the patients, prednisone was withdrawn and they remained on low-dose MTX during clinical remission. Only 30% of the patients continued steroid therapy in low doses (5 mg/per day). The patients began therapy with a starting dose of MTX 8.25 ± 1.39 mg/week (35%—10 mg, 60%—7.5 mg, 5%—5 mg). In 30% of the cases, the dosage of MTX was increased, to a mean value of 12.92 ± 2.24 mg/week. The minimum dosage in clinical remission was 7.13 ± 3.68 mg/week.

The clinical remission rate was 60%. Remission occurred on average after 6.00 ± 4.22 months of treatment. The survival rate was 85%. Skin lesions were observed in 90% of the patients, and affected mucous membranes in 45%. In 20% of the cases, MTX therapy was discontinued after achieving remission, after a mean period of 10.00 ± 5.96 months. MTX therapy was withdrawn in 20% of the cases due to side effects: 10% on account of poor tolerance, 10% during hospitalization for renal failure and pneumonia. For two patients from this group, the therapy was modified to high doses of prednisone monotherapy (40–60 mg/day; no remission has been obtained) because of poor tolerance to MTX.

In total, 55% of the patients presented with an exacerbation of the disease while reducing MTX doses. In addition, 30% of the patients developed side effects, including hair loss, weakness or mucosal lesions in the form of erosions. Laboratory tests conducted during the therapy showed no deviations from the norm. Additionally, 10% of the patients did not take the medications in accordance with the recommendations, with some reporting a one-week break in the treatment connected with infection. As in Group 1, this resulted in an extended time to achieve clinical remission.

### 3.3. Comparison of the Groups

The mean patient age was higher in Group 1 (81.75 ± 10.04 years) than in Group 2 (73.35 ± 11.42 years; *p* < 0.05) (Figure 1A).

The survival rate was 100% in Group 1 and 85% in Group 2 (*p* < 0.01). A slightly better survival rate was noted in the MTX group; in Group 2 the survival rate decreased with the increase in age (*p* < 0.05) (Figure 1B).

No significant differences were found between the groups regarding the time needed to achieve clinical remission: 3.80 ± 3.17 months in Group 1, 6.00 ± 4.22 months in Group 2. Clinical remission was noted in 100% of the cases in Group 1 and 60% in Group 2 (*p* < 0.001) (Figure 1C).

No significant differences were found regarding the starting dosage of MTX (Group 1: 7.63 ± 1.67; Group 2: 8.25 ± 1.39 mg/week). The minimum dosage in clinical remission was lower in Group 1: 4.88 ± 2.32 mg/week (*p* < 0.05). In Group 1, the MTX dosage was reduced with the increase in age (*p* < 0.05). In Group 2, the MTX dosage increased slightly, but not significantly, with the increase in age. In Group 2, MTX was found to increase with decrease in steroid dose (*p* < 0.05) (Figure 1D).

The presence of lesions on mucous membranes was more common in Group 2 (45%) than in Group 1 (5%; *p* < 0.01) (Figure 2A).

MTX therapy was well tolerated by all the patients in Group 1, but in only 80% in Group 2 (*p* < 0.05; Figure 2B). Following remission and the end of therapy, disease recurrence was more common in Group 2 (80%) than in Group 1 (40%; *p* < 0.01) (Figure 2C).

Exacerbations during treatment were more often observed in Group 2 (55%) than in Group 1 (42.1%; *p* < 0.05) (Figure 2D). In both groups, health was found to deteriorate when decreasing the MTX dosage.

Side effects were reported in 30% of the cases in Group 1 and in 35% in Group 2; this difference was not significant. Neither a previous history of internal diseases (diabetes and hypertension) nor medication use (furosemide and neuroleptics) affected the treatment in either group. Laboratory tests conducted during the therapy showed no deviations from the norm. In both groups, none of the patients presented with severe side effects related to MTX treatment (Table 1).

## 4. Discussion

Our findings, based on a retrospective analysis of patient records over a number of years, indicate that MTX in monotherapy is an effective treatment in patients with BP. This is an important finding as the high mortality rate and neurological complications observed in elderly patients with BP have resulted in a pressing need for new forms of treatment [21,22,23,24].

It should be noted that clinical remission was observed in all patients treated by MTX alone (Group 1). In addition, the time required to achieve remission was similar to that observed for the MTX and steroids group (Group 2). Hence, the time needed to achieve remission should not influence the decision to add steroids to MTX treatment. Our findings were not confirmed by those of previous studies; in these cases, either the patients were not divided into groups based on treatment type, or the remission time was not compared [2,3,4,5,6].

An initial dosage of MTX higher than 7.5 mg appears necessary to achieve remission without increasing the dose; 10 mg appears to be appropriate. In our present study, 25% of the patients had to receive increased doses due to lack of clinical remission; more than half of these were given 7.5 mg/week initially.

Previous studies tested the use of a 5 mg/week starting dose of MTX for patients with BP [2,4]; however, the dosage was later increased to a maximum of 12.5 mg/week in order to achieve remission. As MTX therapy seems well tolerated, Kjellman et al. suggest increasing the dosage as a safer alternative to systemic steroid therapy [2]. Other studies have used a starting dosage ranging from 5 to 7.5 mg/week, being increased to a maximum of 10 mg/week in the case of an unsatisfactory response [3]. Interestingly, a recent study found a dose of 10 mg/week to result in clinical remission in all patients with BP [6].

The severity of BP influences the starting dose of MTX. A correct dosage helps achieve remission quicker, which in turn affects the quality of life. All of the patients showed good responses to treatment and high tolerance to MTX therapy with no need to change treatment to systemic steroids.

A dosage above 10 mg appears suitable in the case of a high BPDAI score. In two studies, increasing the dosage of MTX until satisfactory therapeutic effects were achieved resulted in a maximum dosage between 12.5–17.5 mg/week [2,4], with one study noting a correlation between the severity of the disease and the time to remission [2]. It is possible that the need to increase the dosage of MTX in patients with advanced disease resulted in a prolonged time to remission. In our group of patients, the mean starting dose of MTX was 7.63 ± 1.67 mg/week in those treated only with MTX (Group 1), and 8.25 ± 1.39 mg/week in MTX + steroids (Group 2). In Group 1, it was necessary to increase the dosage in 25% of the cases to a mean value of 10.50 ± 2.45 mg/week; in Group 2, 30% of the patients received an increased dose of MTX to a mean value of 12.92 ± 2.24 mg/week.

In our study, due to exacerbations occurring while reducing MTX doses or during fast withdrawal, 85% of the patients were left on a minimal MTX dosage (mean: 4.88 ± 2.32 mg/week) for a period of several months. However, it was not possible to determine the mean time needed to achieve full recovery due to the variation in observation times, nor was it possible to describe the patients’ observations after several years following recovery. Nonetheless, longer low-dosage MTX therapy appears to offer a lower risk of side effects than systemic steroid therapy, despite the perceived faster improvement associated with the latter.

A series of papers have examined the complications of steroid therapy and increased mortality in patients with BP [11,21,22,25]. The risk of developing pulmonary embolism or pneumonia was found to be increased following a diagnosis of BP, and BP itself is associated with high mortality [21,25]. In addition, another study notes a considerable case–fatality rate in patients with BP, and older patients who require a higher dosage of oral glucocorticosteroids during hospital discharge and who have low serum albumin levels are at a greater risk of death within the first year following hospitalization [22]. These prognostic factors should be considered in the care of patients with BP, as well as in the design of future clinical trials.

A study comparing oral and topical steroid therapies indicates numerous complications and side effects caused by steroid treatment. These include biological (diabetes mellitus), radiological (stroke, pneumonia and bone fracture), or bacteriological effects (septicemia, arthritis and peritonitis) [11,12].

The collected data indicate that MTX monotherapy is more commonly used in elderly patients with BP due to the possibility of side effects associated with systemic steroid treatment. This is reflected in the higher mean age of the patients in Group 1. This implies that age is a major factor in choosing BP therapy. The choice of treatment is also connected to the higher survival rate in Group 1 (100%) compared to combined steroids and MTX therapy (Group 2).

This is confirmed by Kjellman et al., who report higher mean age in a group of patients treated with MTX than in those treated with glucocorticosteroid monotherapy. In addition, different mortality values were found between groups treated with MTX, as monotherapy or combined with steroid therapy, and groups treated without MTX, i.e., with topical and oral glucocorticosteroids. The highest survival rate was observed for the group treated with MTX as monotherapy [2]. Unfortunately, it is not possible to compare these findings as previous studies examining the use of MTX for treating BP did not divide the groups based on treatment type. But conclusion of this study is that MTX was the most effective treatment, with only a few adverse effects and a tendency toward better survival rates in patients with moderate to severe disease.

Laboratory tests conducted during the MTX treatment showed no deviations from the norm. None of the patients presented with severe side effects such as bone marrow damage, renal or liver failure.

The low incidence of skin lesions on mucous membranes in patients from Group 1 suggests that MTX was used as monotherapy less frequently for this indication. Conversely, the more common occurrence of BP lesions on mucous membranes in Group 2 suggests that steroid therapy was a more popular option for patients experiencing great discomfort and a decreased quality of life. Patients with mucosal changes are unable to eat, can present with dyspnoea and have trouble talking. In these cases, adding steroids results in a faster improvement. However, the available studies on MTX as monotherapy in patients with BP include little information about the differences in the localization of BP changes.

In Group 2, the mean dosage of prednisone was 21.58 ± 12.68 mg/day. This value was definitely lower than the recommended dosage of prednisone in monotherapy, compatible with the EADV consensus and also the Polish Dermatological Society (40–60 mg) [1]. Moreover, a higher dose of MTX allowed for significantly lower doses of steroids in Group 2 (*p* < 0.05). In order to minimize the possibility of post-steroid side effects that occur with the increasing age of the patients with BP, it was decided to use higher doses of MTX and lower doses of steroids. Similarly, in a previous study, the mean dose of prednisone used in combination with MTX was 27.5 ± 26.5 mg; in this case, the patients were treated with a high dose of prednisone (40–120 mg/d) beforehand. Including MTX into the combined therapy allowed glucocorticosteroid doses to be reduced [3].

In Group 2 (combined treatment), steroids were completely discontinued in 40% of the cases with improvement being maintained on minimal MTX doses, whereas 60% of the patients achieved clinical remission. Two of the patients, without previous remission and with poor tolerance to MTX (nausea, vomiting and erosions in the oral mucosa), did not achieve clinical remission despite modifying the therapy to systemic steroids.

Upon examining the collected findings, it is worth noting that combined steroid and MTX therapy is chosen just as often as MTX in monotherapy. The differences in achieving clinical remission between the groups are associated with the severity of the disease and the presence of advanced BP lesions. Patients with numerous blisters, including lesions on the mucosa, need intensive therapy, and combined MTX and glucocorticosteroid treatment seems to be a good choice. Due to the substantial decrease in quality of life associated with mucosal changes, it is important to achieve remission as quickly as possible; in such cases, combined therapy based on MTX with low doses of prednisone is recommended.

Our findings indicate that patients with BP should be treated with MTX in monotherapy. In addition, dermatologists should focus on reducing systemic steroid use during treatment to reduce the risk of side effects related to steroids and mortality rate. Similar results can be observed in other studies on the effectiveness of MTX in monotherapy in BP. MTX therapy is effective and safe, both when applied alone and with low doses of steroids [2,3,4,5].

In contrast to our findings, Kjellman et al. note a significant correlation between the disease severity and the time to achieve remission [2]. However, this parameter may have an influence on the time needed to achieve remission; as such, further observations of those patients would be advisable.

It should be mentioned that MTX therapy can cause serious complications. As such, certain contraindications must be taken into account when implementing the treatment. These include hypersensitivity to MTX, high stage liver and/or renal disfunction, hematological diseases, acute or chronic infections, ulceration of oral and gastrointestinal mucous membranes, recent surgical procedures, immune deficiencies, pregnancy and breast feeding.

## 5. Limitations of the Study

The study presents an alternative method of treating BP to steroid therapy. However, the study was based on a relatively small group of patients due to our strict inclusion and exclusion criteria. It is intended to follow up these patients more carefully in the future, and to perform a more prospective study to further evaluate the benefit of methotrexate treatment.

## 6. Conclusions

Our analysis compares MTX as monotherapy and as combined treatment with steroids in a group of elderly patients suffering from BP. Our findings indicate that MTX in monotherapy is an effective and safe therapeutic method for elderly patients.

Both treatments yielded comparable time to remission; hence, this is not an important consideration when choosing treatment. As steroids should ensure quicker remission, combined therapy is a more frequent choice when mucosal lesions are present.

The combined MTX + steroid therapy used lower doses of glucocorticosteroids than recommended in steroid monotherapy in BP patients. The history of internal diseases did not affect the treatment for either tested regimen. A higher mortality rate was observed in the group treated with steroids; this may be related to the steroid-associated side effects and prolonged use of medication.

## Figures and Tables

**Figure 1 jcm-12-01638-f001:**
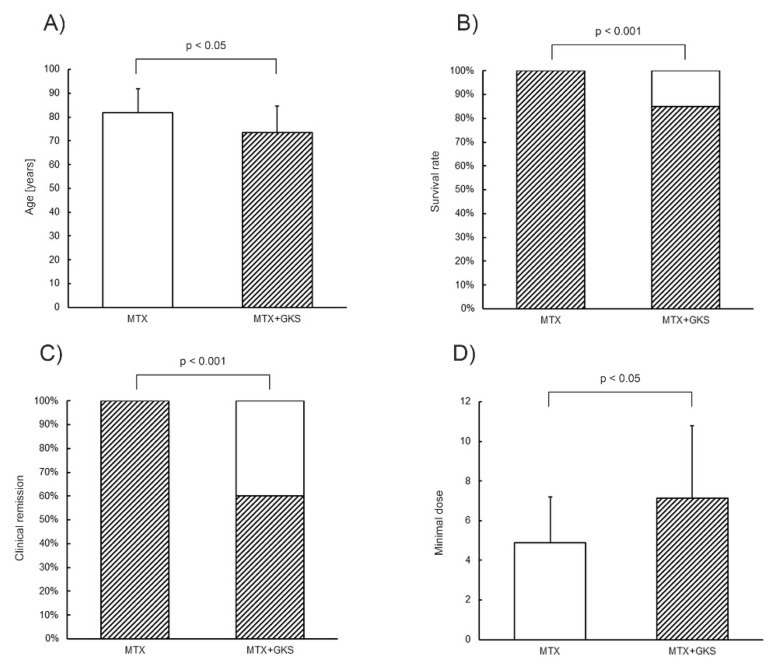
Comparison of Group 1 and Group 2 with regard to (**A**) age, (**B**) survival rate, (**C**) clinical remission, (**D**) minimal dose. MTX—Methotrexate, GKS—glucocorticosteroids. (**A**,**D**) White bar—group 1, shadow bar—group 2. The results of semiquantitative analysis are expressed as the mean ± standard deviation. The level of significance is defined where *p* < 0.05. (**B**,**C**) White and shadow bar—presented feature in the group in %.

**Figure 2 jcm-12-01638-f002:**
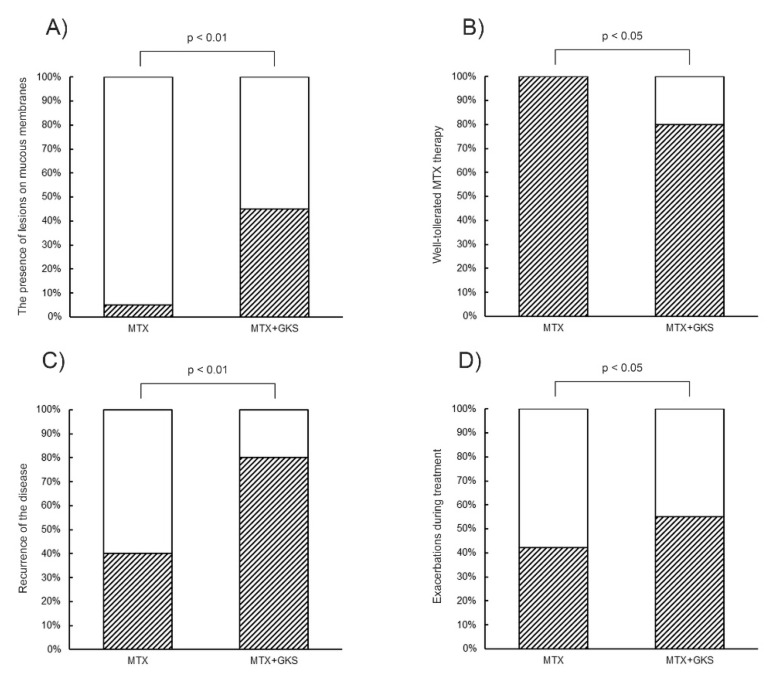
Comparison between Group 1 and Group 2 with regard to (**A**) the presence of lesions on mucosa membranes, (**B**) well tolerated MTX therapy, (**C**) recurrence of the disease, (**D**) exacerbations during treatment. MTX—Methotrexate, GKS—glucocorticosteroids. The results of semiquantitative analysis are expressed as the mean ± standard deviation. The level of significance is defined where *p* < 0.05. White and shadow bar—presented feature in the group in %.

**Table 1 jcm-12-01638-t001:** Characteristic of the groups.

	Number of Patients	Average Age [Years]	Starting Dosage of MTX [mg/wk]	Range of MTX [mg]	Min. Dosage of MTX	Clinical Remission	Time to Remission [Months]	Survival Rate [%]	Withdrawn Therapy [%]
MTX group 1	28	81.75 ± 10.04	7.63 ± 1.67	5–10	4.88 ± 2.32	100%	3.80 ± 3.17	100	0
MTX + GKS group 2	31	73.35 ± 11.42	8.25 ± 1.39	5–10	7.13 ± 3.68	60%	6.00 ± 4.22	85	20

## Data Availability

Not applicable.

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
