# Peer review of "Can Methotrexate Be Employed as Monotherapy for Bullous Pemphigoid? Analysis of Efficiency and Tolerance of Methotrexate Treatment in Patients with Bullous Pemphigoid"

_jcm, 2023, doi:10.3390/jcm12041638_

Round 1

Reviewer 1 Report

1. The use of methotrexate as a safe drug with similar efficacy to oral corticosteroids

2. The theme is original and provides the use of methotrexate in this pathology of elderly patients 3. There are very few publications on the use of methotrexate compared to corticosteroids in this pathology. 4. It is methodologically correct.

It has a not very high number of patients
It will need in the future of new studies

5. The  conclusions are correct 6. The bibliography is appropriate. 7. The figures and tables are adequate and of an adequate presentation.

Author Response

----------------------------------------------------Department of Dermatology----------------------------------------------

Medical University of Lodz

Hallera 1 sq., 94-049 Lodz, Poland

tel. (48-42) 68 67 981

fax. (48-42) 68 84 565

                                                                                                             2ndof February 2023

Dear All,

We are very grateful for the remarks of the Reviewers. There is no doubt that these reviews helped us to improve our paper. All remarks were applied to the new version of the manuscript and requested corrections were made.

Below we have referred to all the comments of the Reviewer #1:

1.The use of methotrexate as a safe drug with similar efficacy to oral corticosteroids

2.The theme is original and provides the use of methotrexate in this pathology of elderly patients

3.There are very few publications on the use of methotrexate compared to corticosteroids in this pathology.

4.It is methodologically correct.

  1. The conclusions are correct

6.The bibliography is appropriate.

  1. The figures and tables are adequate and of an adequate presentation.

It has a not very high number of patients. It will need in the future of new studies

Thank you for this review. We tried to present an alternative and effective method of treatment to steroid therapy. We had a larger group of patients at the beginning, but the inclusion and exclusion criteria limited it. Of course, we plan further research and evaluation of this treatment.

Reviewer 2 Report

The clinical trials of the drug MTX were well planned. However, it was worth excluding cancer and tuberculosis in the group of examined patients. However, this does not affect the very high rating of this publication. It will certainly be of interest to clinicians.

Author Response

Dear All,

We are very grateful for the remarks of the Reviewers. There is no doubt that these reviews helped us to improve our paper. All remarks were applied to the new version of the manuscript and requested corrections were made.

Below we have referred to all the comments of the Reviewer #2:

The clinical trials of the drug MTX were well planned. However, it was worth excluding cancer and tuberculosis in the group of examined patients. However, this does not affect the very high rating of this publication. It will certainly be of interest to clinicians.

We did not mention this in our study, but in accordance with the principles and guidelines for the treatment of bullous diseases, tuberculosis was excluded in each patient. And all BP patients are screened for cancer before immunosuppressive treatment.

Reviewer 3 Report

Comments:

-        Extensive editing of English language and style is required.

-        Researcher retrospectively compare between methotrexate monotherapy and combination of methotrexate and systemic steroid therapy in patients with Bullous pemphigoid (BP)

Introduction

1.     Please provide evidence to support the use of MTX in BP treatment since the cited references (number 13-20) mostly referred to MTX in rheumatoid arthritis (page 2, line 71-82).

Methods

1.     Clearer inclusion and exclusion criteria should be identified. Some points seemed to be conflicted, as the exclusion criteria stated that “Patients who did not achieved treatment compliance and stopped taking methotrexate were excluded from the study.” However, in the results reported that “Additionally, 10% of the patients did not take the medications in 163 accordance with the recommendations and therefore…”. ïƒ  this 10% population should be initially excluded (or not?)

2.     All outcomes in the study should be mentioned in methods. In addition, please explain how did you calculate survival rate, and other related outcomes?

Results

1.     The baseline characteristics of patients should be arranged in tabular form to explicitly compare between the two treatment groups. 

2.     Figure 1C, 1D, and Figure 2 have not been mentioned in the text. What is minimal dose (in Figure 1D)? 

3.     How do you justify association between diseases and MTX therapy “History of internal diseases such as hypertension or diabetes as well as treatment with furosemide or neuroleptics did not influence the MTX therapy.”?

4.     How were “patients presented with an exacerbation of the disease during attempts to reduce MTX doses” managed?

5.     Please be consistent when presenting the number format in the text (comma vs dot).

6.     “In group 1 MTX therapy was well-tolerated by all the patients, in group 2 in only 80% of the cases (p<0.05)” How can “well-tolerated” be define in this scenario? 

Discussion

1.     Page 6, Line 245, reference number [2.4], did you mean [2-4] ?  

2.     Page 7, Line 297-298 “It is also connected to the mortality rate–100% in group 1, which was higher than in the group treated with combined 298 steroids and MTX therapy (group 2).” ïƒ  Is this sentence correct? Why does in the abstract showed “The mortality rate was higher in the group treated with steroids. (Page1, line 18-19)” and in conclusion section showed “Higher mortality rate in the group treated with steroids may be related to the steroid-assiociated side effects and prolonged use of the medications.”. (assiociated ïƒ  typo)

3.     Please explain if there is any limitation of this study and future suggestion.

Author Response

Dear All,

We are very grateful for the remarks of the Reviewers. There is no doubt that these reviews helped us to improve our paper. All remarks were applied to the new version of the manuscript and requested corrections were made.

Below we have referred to all the comments of the Reviewer #3:

Comments:

Extensive editing of English language and style is required.

The English language has been improved according to the comments of both reviewers. We have corrected the entire manuscript. We have removed typographical errors from the text.

Researcher retrospectively compare between methotrexate monotherapy and combination of methotrexate and systemic steroid therapy in patients with Bullous pemphigoid (BP). 

Introduction

1.Please provide evidence to support the use of MTX in BP treatment since the cited references (number 13-20) mostly referred to MTX in rheumatoid arthritis (page 2, line 71-82).

We added a proper publications to the manuscript.  As below a single papers described therapy with MTX in BP.

Paul MA, Jorizzo JL, Fleischer AB Jr, White WL. Low-dose methotrexate treatment in elderly patients with bullous pemphigoid. J Am Acad Dermatol. 1994 Oct;31(4):620-5. doi: 10.1016/s0190-9622(94)70227-6. PMID: 8089289.

Kolla, A, Shah, P, Cymerman, R, Fruchter, R, Adotama, P, Soter, NA. Assessing the use of methotrexate as an alternate therapy for pemphigus vulgaris and pemphigus foliaceus. Dermatologic Therapy. 2022; 35( 8):e15661. doi:10.1111/dth.15661

A Fisch, L Morin, T Talme, K Johnell, I Gallais Sérézal Low-Dose Methotrexate Use and Safety for Older Patients With Bullous Pemphigoid and Impaired Renal Function: A Cohort Study J Am Acad Dermatol 2020 Feb 20

Methods

1.Clearer inclusion and exclusion criteria should be identified. Some points seemed to be conflicted, as the exclusion criteria stated that “Patients who did not achieved treatment compliance and stopped taking methotrexate were excluded from the study.”

However, in the results reported that “Additionally, 10% of the patients did not take the medications in 163 accordance with the recommendations and therefore…”.this 10% population should be initially excluded (or not?)

Initially, 120 patients with bullous pemphigoid diagnosed on the basis of immunopathological examination were enrolled in the study. Patients who did not achieved treatment compliance and/or stopped taking methotrexate were excluded from the study.

The cause of stopping taking MTX was high level of liver tests and severe anemia after first or second dose of MTX. We added this information to the manuscript.

Patients who did not achieved treatment compliance – it is a group of patients who have no possibility to come to our clinic for every visit and the visit were conducted for example in GP center.

The group of 10% BP patients which were added to study group was at least 6 months in observation. But in this group was observed for example one week break in treatment connected with infection.

2.All outcomes in the study should be mentioned in methods. In addition, please explain how did you calculate survival rate, and other related outcomes?

We added information to manuscript. In material and methods section.

Relative survival of a disease, in survival analysis, was calculated by dividing the overall survival after diagnosis by the survival as observed in a similar population not diagnosed with that disease. A similar population is composed of individuals with at least age and gender similar to those diagnosed with the disease.

Mortality ratio - the ratio of the number of deaths in a given population to the number of deaths expected, a form of indirectly standardized rates, where the categories are usually "defined by age, gender and race or ethnicity” MR is described as a proportional comparison to the numbers of deaths that would have been expected if the population had been of a standard composition in terms of age, gender.

Results

1.The baseline characteristics of patients should be arranged in tabular form to explicitly compare between the two treatment groups. 

Thank you for this suggestion. We did not want to repeat the same data in the text and in the table. We don't know what to do with the other three reviewers' suggestions that this section is well presented in the text.

  1. Figure 1C, 1D, and Figure 2 have not been mentioned in the text. What is minimal dose (in Figure 1D)? 

We added mention of these figures in the text.

As it is mentioned in the text: The minimum dose of MTX in clinical remission was 4.88± 2.32 mg/wk (range 2-10 mg).

3.How do you justify association between diseases and MTX therapy “History of internal diseases such as hypertension or diabetes as well as treatment with furosemide or neuroleptics did not influence the MTX therapy.”?

We did not observe an increase in the incidence of side effects related to comorbidities and medications. We also excluded provocation of BP lesions by drugs such as furosemide and neuroleptics.

4.How were “patients presented with an exacerbation of the disease during attempts to reduce MTX doses” managed?

The dose of methotrexate was increased to the earlier dose that controlled the disease. No additional treatment was required.

5.Please be consistent when presenting the number format in the text (comma vs dot).

It has been corrected in the text (page 5 line 196, line 210, page 7 line 268).

6.“In group 1 MTX therapy was well-tolerated by all the patients, in group 2 in only 80% of the cases (p<0.05)” How can “well-tolerated” be define in this scenario? 

No side effects that could lead to discontinuation of the drug were observed.

Discussion

Page 6, Line 245, reference number [2.4], did you mean [2-4] ?  

The correct format is: [2,4]. It has been corrected in the text.

Page 7, Line 297-298 “It is also connected to the mortality rate–100% in group 1, which was higher than in the group treated with combined 298 steroids and MTX therapy (group 2).” à Is this sentence correct? Why does in the abstract showed “The mortality rate was higher in the group treated with steroids. (Page1, line 18-19)” and in conclusion section showed “Higher mortality rate in the group treated with steroids may be related to the steroid-assiociated side effects and prolonged use of the medications.”. (assiociated à typo)

The correct form of the sentence is: 298 “It is also connected to the survival rate–100% in group 1, which was higher than in the group treated with combined 298 steroids and MTX therapy (group 2).”

  1. Please explain if there is any limitation of this study and future suggestion.

We tried to present an alternative and effective method of treatment to steroid therapy. We had a larger group of patients at the beginning, but the inclusion and exclusion criteria limited it. To further evaluate the benefit of methotrexate treatment, we are now designe the study to be able to follow up these patients more carefully and to conduct a prospective study that will give further information.

                                                                                                      Kind regards

                                                             Agnieszka Å»ebrowska

Reviewer 4 Report

One of the biggest concerns during the treatment of BP is the use of MTX because the disease is more prevalent in elderly patients.

You have not described moderate to severe adverse effects during therapy with doses of up to 15 mg of MTX per week (in the group, mean age of patients was 81 years, plus or minus 10 years). How long were patients observed for adverse effects?

Author Response

Dear All,

We are very grateful for the remarks of the Reviewers. There is no doubt that these reviews helped us to improve our paper. All remarks were applied to the new version of the manuscript and requested corrections were made.

Below we have referred to all the comments of the Reviewer #4:

One of the biggest concerns during the treatment of BP is the use of MTX because the disease is more prevalent in elderly patients.

You have not described moderate to severe adverse effects during therapy with doses of up to 15 mg of MTX per week (in the group, mean age of patients was 81 years, plus or minus 10 years). How long were patients observed for adverse effects?

MTX may cause severe side effects including bone marrow damage, renal or liver dysfunction or neurotoxic symptoms. Mild or moderate side effects like increased risk of infection, gastrointestinal symptoms, hair loss, weakness, fatigue and skin rashes are described.

Initially, 120 patients with bullous pemphigoid diagnosed on the basis of immunopathological examination were enrolled in the study. Patients who did not achieved treatment compliance and/or stopped taking methotrexate were excluded from the study.

The cause of stopping taking MTX was high level of liver tests and severe anemia. We added this information to the manuscript.

Finally a study group consists of 40 BP patients. In our study group (described in result section) there were only mild side effects.

“In group 1, the MTX therapy was well tolerated in 100% of the cases and there was no need to withdraw MTX on account of the side effects. The main complaints reported by the patients were 5% hair loss, 10% weakness and fatigue. Laboratory tests (LFTs, morphology) conducted during the therapy showed no deviations from the norm.

Patients were monitored for adverse events throughout the treatment period and after treatment discontinuation.

The shortest follow-up time was at least 6 months, but if it lasted longer, the follow-up was extended.
